

# Label-Guided relation prototype generation for Continual Relation Extraction

Shuang Liu[1], XunQin Chen[1], Peng Chen[2] and Simon Kolmanič[3]

[1] School of Computer Science and Engineering, Dalian Minzu University, Dalian, China
[2] School of Computer and Software, Dalian Neusoft University of Information, Dalian, China
[3] Faculty of Electrical Engineering and Computer Science, University of Maribor, Maribor, Slovenia

## ABSTRACT

Continual relation extraction (CRE) aims to extract relations towards the continuous and iterative arrival of new data. To address the problem of catastrophic forgetting, some existing research endeavors have focused on exploring memory replay methods by storing typical historical learned instances or embedding all observed relations as prototypes by averaging the hidden representation of samples and replaying them in the subsequent training process. However, this prototype generation method overlooks the rich semantic information within the label namespace and are also constrained by memory size, resulting in inadequate descriptions of relation semantics by relation prototypes. To this end, we introduce an approach termed Label-Guided Relation Prototype Generation. Initially, we enhance the representations of label embeddings through a technique named label knowledge infusion. Following that, we utilize the multi-head attention mechanism to form relation prototypes, allowing them to capture diverse aspects of typical instances. The embeddings of relation labels are utilized at this stage, leveraging their contained semantics. Additionally, we propose a feature-based distillation loss function called multi-similarity distillation, to ensure the model retains prior knowledge after learning new tasks. The experimental results indicate that our method has achieved competitive performance compared to the state-of-the-art baseline models in CRE.

## INTRODUCTION

Continual relation extraction (CRE) (*Wang et al., 2019b*) requires models to learn new relations from a class-incremental data stream.

However, as new tasks are learned, and new relations continuously emerge, the model tends to forget knowledge about old relations, a phenomenon known as catastrophic forgetting.

Some recent works have proposed a variety of methods to alleviate the catastrophic forgetting problem in continual learning, including regularization methods (*Liu et al., 2018*), dynamic architecture methods (*Fernando et al., 2017*) and memory-based (*Chaudhry et al., 2018*) methods. In recent years, the previous works have demonstrated

Corresponding author
Peng Chen, chenpeng@neusoft.edu.cn

the effectiveness of memory-based methods in CRE. The main idea of the memory-based methods is to mitigate catastrophic forgetting by replaying samples stored in memory after learning new tasks. Due to constraints in memory and computational resources, these approaches typically store only typical samples from the dataset. The task of memory replay is to fully leverage the information contained in these instances to maintain the knowledge acquired in previous learning tasks. Relation prototypes to enhance continual relation extraction (RP-CRE) (*Cui et al., 2021*) devises a relation prototype-based method to refine the samples embedding through a memory net to fully utilize typical samples, while the relation prototypes calculation is relatively simplistic and sensitive to the typical samples. Consistent representation learning (CRL) (*Zhao et al., 2022*) introduces supervised contrastive learning and knowledge distillation to maintain prior knowledge. While it narrows the representations of samples within the same relation using supervised contrastive learning, it falls short in maintaining a sufficient separation between the representations of samples from different relations, potentially leading to confusion. Continual Relation Extraction framework with Contrastive Learning (CRECL) (*Hu et al., 2022*) comprises a classification network and a contrastive network to decouple the representation and classification processes. However, its method for generating relation prototypes is still limited to the instances stored in memory.

Overall, existing memory-based methods may encounter follow limitations in CRE, restricting the model's performance.

- The relation prototype calculation in existing memory-based CRE methods is sensitive to typical samples. Typically, these methods generate relation prototypes by directly averaging the vector representations of typical samples. If the quality of the selected typical samples is low, the obtained expression of relation prototypes may not be robust enough.
- These methods generally employ contrastive learning with relation prototypes serving as anchor points for memory replay or knowledge distillation. However, this kind of approach do not effectively preserve a notable distinction between the representations of samples belonging to different relations and also may lead to the loss of specific features associated with certain samples, and this approach also blurs the distinct features of instances.

To address the above challenges, we propose an approach named Label-Guided Relation Prototype Generation. Specifically, we utilize a multi-head attention mechanism to combine the hidden representations of instances, so that varying importance of instances can be introduced in the prototypes. We use the embedding of relation labels as query vectors for the attention module, leveraging the rich semantic information they contained (*Chen et al., 2022*). However, labels are typically highly abstract representations of relations, providing limited specific details about the relationships. To capture patterns among different samples within the same relation, we concatenate labels and text as input to the encoder while learning the new task, with the expectation that more specific semantic knowledge can be infused into the relation labels. After replaying the memory, we also design a multi-similarity distillation loss function for knowledge distillation. By mining

hard positive and hard negative samples, we aim to ensure that the embedding space remains undamaged during the learning of new tasks while making instances between different relationship categories as distinguishable as possible.

In summary, our contributions in this paper are summarized as follows:

- We propose a novel method for generating relation prototypes, using knowledge-injected relation tokens to capture distinctive features of different typical instances.
- We designed a distillation loss function based on typical instances. This aims to maintain old knowledge while making instances belonging to different relations more distinguishable.
- The comprehensive experiments on two benchmark datasets empirically demonstrate that our proposed method achieves competitive performance compared to the state-of-the-art baseline models.

## RELATED WORK

Existing continual learning models mainly focus on three areas: (1) Regularization-based methods (*Kirkpatrick et al., 2017*; *Liu et al., 2018*) aim to strike a balance between learning new information and retaining knowledge from previous tasks by introducing constraints or penalties during the training process, limiting the extent to which the model can change its parameters. (2) Dynamic architecture methods (*Chen, Goodfellow & Shlens, 2015*; *Fernando et al., 2017*) provide a flexible framework that allows the model to adapt to new tasks without erasing previously acquired knowledge. These approaches attempt to overcome catastrophic forgetting by dynamically adjusting the model's structure or using mechanisms that selectively focus on relevant information for each task. (3) Memory-based methods (*Wang et al., 2019b*; *Dong et al., 2021*) equip models with mechanisms to store and retrieve information effectively. ERDIL (*Dong et al., 2021*) leveraging exemplar relation graphs and a specialized loss function to effectively handle few-shot class-incremental learning tasks. EA-EMR (*Wang et al., 2019b*) is a classical CRE model. The proposed methods includes a modified stochastic gradient approach and an explicit alignment model.

In recent studies, memory-based methods have been extensively applied in the CRE scenarios. The results of these studies also indicate the effectiveness of memory-based approaches in this domain. Additionally, the prototype method (*Cui et al., 2021*; *Zhao et al., 2022*) is increasingly being adopted in recent research to preserve existing knowledge and reduce overfitting.

## METHODOLOGY

### Problem formulation

In continual relation extraction, there is a series of K tasks $\{T_1, T_2, \ldots, T_k\}$, where the k-th task has its own training set $D_k$ and relation set $R_k$. The relation sets of different tasks are disjoint. Each instance in the training set $D_k$ is represented as $(x_i, r_i)$, where $x_i$ is the input text, and $r_i$ is the relation label. The objective of continual relation learning is to avoid catastrophic forgetting of previous learning tasks while learning new tasks, ensuring that

satisfactory performance can still be achieved within the relation set $\hat{R}_k$, where $\hat{R}_k = \bigcup_{i=1}^{k} R_i$ is the relation set already observed till the k-th task.

Additionally, an episodic memory space $\hat{M}_k = \bigcup_{i=1}^{k} M_i$ will be used to store some typical samples of previously learned data for memory replay. Here, $M_i$ is selected from the training set $D_i$ of the i-th task. Constrained by memory size and resource consumption, storing all observed data is impractical, so typically, only the most representative $m$ instances are stored.

## Framework

The overall framework is shown in Fig. 1. For a new task $T_k$, our proposed method mainly includes four stages, with the specific training process shown in Algorithm 1. (1) New task training and label knowledge injection (Line 2~10): The purpose of this stage is to learn the newly emerging relations $R_k$ in $T_k$. Relation labels will be inserted into the text at this stage to capture specific descriptions of the relationships in the text. Additionally, for each $r$ in $R_k$, the K-means algorithm will be used to select representative instances from $D_k$ to be stored in memory $\hat{M}_k$. (2) Relation prototype generation (Line 11): The embeddings of relation label $r$ and the instances stored in memory are used as inputs for multi-head attention to generate relationship prototypes $h_r$ for each observable relationship type. (3) Memory replay (Line 12~16): Contrastive learning is performed using the classic instances and the generated relationship prototypes to maintain the knowledge previously learned. (4) Multi-similarity distillation (Line 17~21): We conduct distillation learning using the embedding space based on the outputs of the encoder from the previous stage and the current encoder.

## New task training and label knowledge Infusion

For a new task $T_k$, we train on its training set $D_k$. Given an input sentence $x_i$ and two entityes within the sentence, we insert special markers [E11]/[E12] and [E21]/[E22] at the beginning and end of the head and tail entities, respectively, to indicate their positions in the text.

In this stage, we aim to enrich the semantic information within the relation labels. Specifically, we concatenate the relation label with the sentence using the token [SEP]. Formally, for an instance $x_i$ and its corresponding relation label $r_i$, the sequence fed into the encoder takes the form: $[CLS]r_i[SEP]x_i[SEP]$. Although the model's objective is to predict the semantic relation between given entity pairs, the knowledge gained during this process shapes the representations of label tokens. Consequently, the model learns to generate embeddings for label tokens that capture semantic information based on their contextual usage in a sentence. To prevent the model from solely relying on the label and neglecting the context sentence, we employ a random inserting approach. This means that the relation label will be inserted into a sentence with a certain probability.

When learning a new task, for each training instance, we use the embeddings of [E11] and [E21] as representations for the head and tail entities. The input of the classifier is calculated using:

$$h = W_1[h^{11}; h^{21}] + b \tag{1}$$
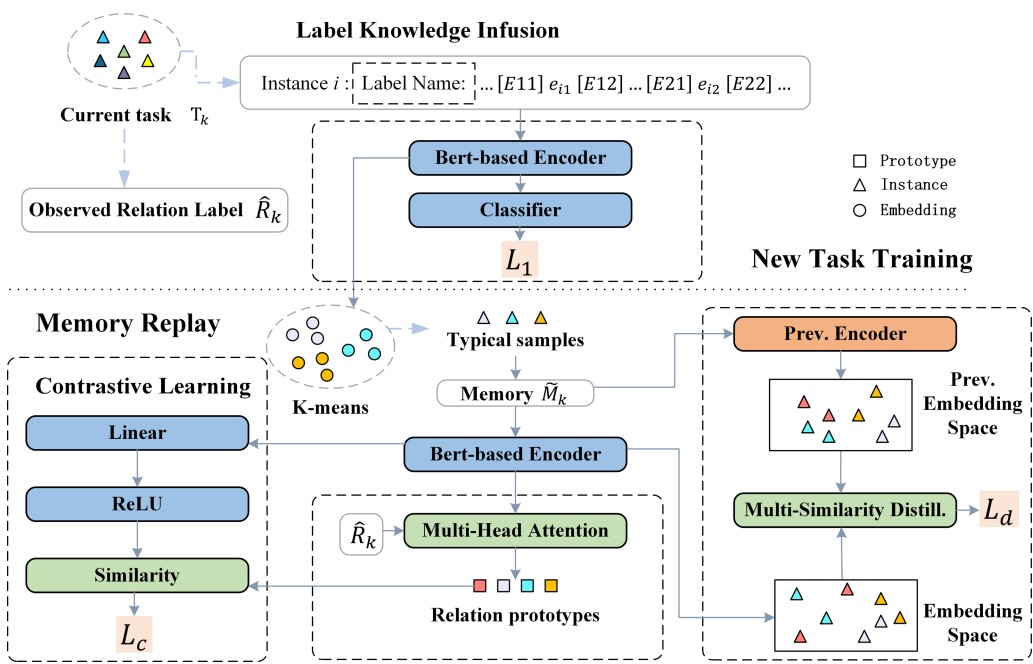

**Figure 1** Framework of the proposed model for task Tk.

where $h^{11}, h^{21} \in \mathbb{R}^d$ are the hidden representations of the tokens [E11], [E21] respectively. $d$ is the dimension of the hidden embeddings output from encoder. [;] denotes the concatenation operation. $W_1 \in \mathbb{R}^{d \times 2d}$ and $b \in \mathbb{R}^d$ are two trainable parameters.

We calculate the probability distribution of the current instance being classified into each relation using the *softmax* function:

$$P(x; \theta_k) = softmax(W_2 h). \tag{2}$$

Here, $W_2 \in \mathbb{R}^{|\tilde{R}_k| \times d}$ is the trainable parameter in the classifier. The loss function for the current learning task is computed as follows:

$$L_1 = -\frac{1}{|D_k|} \sum_{x_i \in D_k} \sum_{r \in R_k} \delta_{r_i, r} \log P(r|x_i; \theta_k). \tag{3}$$

$P(r|x_i; \theta_k)$ represents the probability that the current model $\theta_k$ classifies the input $x_i$ as relation r. $r_i$ is the label of $x_i$ such that if $r = r_i$, $\delta_{r_i, r} = 1$, and 0 otherwise.

### Prototype generation

To fully leverage the semantic knowledge embedded in relation labels, we use the labels to guide the generation of relation prototypes. Specifically, for a particular relation type $r$, its hidden embedding representation $h_r$ is obtained by averaging the encoder outputs of each word in the relation label. For instance, for the relation label "Organization founded", the embedding representation is calculated as follows: $h_{org:founded} = \frac{1}{2}(h_{Organization} + h_{founded})$.

In contrast to the previous approach, which simply averaged the embedding representations of instances stored in memory to obtain relation prototypes, we aim

---

**Algorithm 1** Training procedure for $T_k$

---

**Input:**

The training set of $D_k$ of the k-th task, previous encoder $E_{k-1}$, history memory $\hat{M}_{k-1}$, current relation set $R_k$, history relation set $\hat{R}_{k-1}$.

**Output:**

Encoder $E_k$, memory $\hat{M}_k$, relation set $\hat{R}_k$.

1: $E_k \leftarrow E_{k-1}$;
2: **for** $i \leftarrow 1$ to $epoch_1$ **do**
3:     **for** each $(x, r) \in D_k$ **do**
4:         Insert the relation label $r$ into the text $x$ with a certain probability $p$;
5:         Update $E_k$ with $L_1$;
6:     **end for**
7: **end for**
8: Select informative examples from $D_k$ to store into $M_k$;
9: $\hat{M}_k \leftarrow \hat{M}_{k-1} \cup M_k$;
10: $\hat{R}_k \leftarrow \hat{R}_{k-1} \cup R_k$;
11: Generation prototypes $h_r$ for each $r \in \hat{R}_k$;
12: **for** $i \leftarrow 1$ to $epoch_2$ **do**
13:     **for** each $(x, r) \in \hat{M}_k$ **do**
14:         Update $E_k$ with $L_C$;
15:     **end for**
16: **end for**
17: **for** $i \leftarrow 1$ to $epoch_3$ **do**
18:     **for** each $(x, r) \in \hat{M}_k$ **do**
19:         Update $E_k$ with $L_C$;
20:     **end for**
21: **end for**
22: **return** $E_k, \hat{M}_k, \hat{R}_k$;

---

to make relation prototypes attentive to different patterns within instances. Inspired by KIP-framework (*Zhang et al., 2022*), we utilize the hidden representation of the relation label as the query vector and employ a multi-head attention mechanism to form relation prototypes.

Specifically, we first obtain the hidden embedding representations of instances in memory with relation class $r$ through encoder:

$$H_{x_r} = \{h_x | \forall x \in M_r\} \tag{4}$$

where $M_r$ represents all instances in memory with relation class $r$, and $h_x \in \mathbb{R}^d$ is the embedding representation of instance $x$. The output of the i-th attention head in the multi-head operation is denoted as $p_i \in \mathbb{R}^d$ and is computed as follows:

$$Attention(Q, K, V) = softmax\left(\frac{QK^T}{\sqrt{d}}\right)V \tag{5}$$

$$p_i = Attention\left(h_r W_i^Q, H_{x_r} W_i^K, H_{x_r} W_i^V\right) \tag{6}$$

where $h\_r$ is the hidden embedding corresponding to the relationship label $r$, and $W_i^Q, W_i^K, W_i^V \in \mathbb{R}^{d \times d_h}$, where $d_h = d/N$, with $N$ being the number of attention heads.

At last, the prototype of relation $r$ is calculated by (6):

$$p_r = LN\left(W_3\left[h_1; h_2; \ldots; h_N\right]\right) \tag{7}$$

where $LN(\cdot)$ is the operation of layer normalization, [;] denotes the concatenation operation. $W_3 \in \mathbb{R}^{d \times N d_h}$ is a trainable matrix. $N$ is the number of the attention head. $p_r$ is the prototype of relation $r$.

## Contrastive replay

After learning a new task, to mitigate the problem of catastrophic forgetting caused by changes in model parameters, we need to replay the stored typical instances in memory. In this phase, we employ the contrastive learning for memory replay. Specifically, we use the InfoNCE loss to train the model. For a specific instance $x_i$, the loss is computed as follows:

$$L_c = -\frac{1}{|\widehat{M_k}|} \sum_{x_i \in \widehat{M_k}} log \frac{\exp\left(h_{x_i} \cdot p_{r_i}/\tau_1\right)}{\sum_{r \in \widehat{R_k}} \exp\left(h_{x_i} \cdot p_r/\tau_1\right)} \tag{8}$$

where $r_i$ is the relation label of instance $x_i$, $p_{r_i}$ is the corresponding relation prototype, and $\tau_1$ is the temperature parameter. Utilizing contrastive learning methods provides a more effective classification space, preserving previously learned knowledge while incorporating new knowledge from the current task. However, contrastive replay may alter the distribution of instance's representation. To address this drawback, we propose a multi-similarity distillation method.

## Multi-similarity distillation

In certain previous works, distillation is employed during the memory replay process to preserve prior knowledge. For instance, CRL use the similarity metric between relations in memory as memory knowledge. However, knowledge distillation based on relation prototype may lead to the loss of specific samples of individual instances. In this step, our goal is to ensure that the current model produces features similar to those of the previous model while keeping the model's classification space as clear as possible to avoid confusion.

Considering a specific sample $x_i$, with the current model and the previous model's output hidden representations denoted as $h_{x_i}^l$ and $h_{x_i}^{l-1}$, respectively, we aim to make these two representations as similar as possible. Then, the first term of distillation loss function is computed as follows:

$$L_{nd} = -\frac{1}{|\widehat{M_k}|} \sum_{x_i \in \hat{M}_k} (1 - \cos(h_x^l, h_x^{l-1})) \tag{9}$$

where $\cos(\cdot)$ represents the cosine similarity between two vectors.

Furthermore, to enhance the discriminability of hidden representations of instances of different classes, we employ a contrastive learning approach. Inspired by the multi-similarity

loss (*Wang et al., 2019a*), we introduce a multi-similarity distillation method. We define the similarity between instance $x_i$ and $x_j$ as $S_{ij}$:

$$S_{ij} = h_{x_i}^T h_{x_j}.$$ (10)

Subsequently, we leverage this similarity to mine hard positive and negative samples. The selection criteria for negative sample pairs $\left\{x_i, x_j^-\right\}$ is:

$$S_{ij}^- > \min_{r_k=r_i} S_{ik} - \varepsilon.$$ (11)

Similarly, the selection criteria for positive sample pairs $\left\{x_i, x_j^+\right\}$ is:

$$S_{ij}^+ > \min_{r_k \neq r_i} S_{ik} + \varepsilon.$$ (12)

Here, $\epsilon$ is a hyperparameter. The set containing all selected positive sample pairs is denoted as $N_i$, and the set of negative pairs is denoted as $P_i$. By doing so, we filter out sample pairs with less informative content, thus improving the efficiency of the model. Then we apply a soft weighting to each sample pair based on their similarity. The calculation of weights of positive pairs $w_{ij}^+$ and negative pairs $w_{ij}^-$ is as follows:

$$w_{ij}^- = \frac{1}{e^{\beta(\lambda - S_{ij})} + \sum_{k \in N_i} e^{\beta(S_{ik} - S_{ij})}}$$ (13)

$$w_{ij}^+ = \frac{1}{e^{-\alpha(\lambda - S_{ij})} + \sum_{K \in P_i} e^{-\alpha(S_{ik} - S_{ij})}}$$ (14)

where $\alpha, \beta, \lambda$ are hyper-parameters At last, the second term of distillation loss function is computed as:

$$L_{ms} = -\frac{1}{|\widehat{M_k}|} \sum_{x_i \in \widehat{M_k}} \left( \begin{array}{l} \frac{1}{\alpha} \log\left[1 + \sum_{k \in P_i} e^{-\alpha(S_{ik} - \lambda)}\right] \\ + \frac{1}{\beta} \log\left[1 + \sum_{k \in N_i} e^{\beta(S_{ik} - \lambda)}\right] \end{array} \right).$$ (15)

Compared to methods that randomly select sample pairs or choose the most challenging sample pairs, the multi-similarity distillation approach, when weighting sample pairs, considers not only the similarity of the sample pairs themselves but also tasks into account other sample pairs. This assigns more weight to sample pairs that carry more information, making the method more robust. Through this approach, the output of the current model not only aligns with that of the previous model but also leads to the aggregation of instances of the same category in the embedding space, making it easier to distinguish instances of different categories.

Overall, the distillation loss function is defined as:

$$L_d = L_{nd} + L_{ms}.$$ (16)

# EXPERIMENT

## Datasets

In our experiment, we utilized two benchmark datasets, FewREL (https://github.com/thunlp/FewRel) and TACRED (https://catalog.ldc.upenn.edu/LDC2018T24).

The FewREL (*Han et al., 2018*) datasets is a dataset for few-shot relation classification. To maintain consistency with previous benchmark models, we employed its version with 80 relations, each consisting of 700 instances.

The TACRED (*Zhang et al., 2017*) dataset is a large-scale relation extraction dataset comprising 42 different relations and a total of 106,264 instances. To ensure a balanced distribution of instances for each relation category, consistent with previous benchmark models, the number of training samples for each relation is restricted to 320, and the number of test samples for each relation is limited to 40.

## Experimental setup and evaluation metrics

For a fair comparison, we employ the same setting and obtain the divided data from the open-source code of [5], [6] to guarantee exactly the same task sequence. We report the average accuracy of five different sampling task sequences. The number of stored instances in the memory for each relation is 10 for all methods.

## Baseline

We compared the experimental results with the baseline models listed below.

Embedding aligned EMR (EA-EMR) (*Wang et al., 2019b*) enhances adaptability and knowledge retention amidst changing data and relations by anchoring sentence embeddings and using an explicit alignment model.

Episodic memory activation and reconsolidation (EMAR) (*Han et al., 2020*) activates episodic memory and reconsolidate with relation prototypes during both new and memorized data learning.

Curriculum-meta learning (CML) (*Wu et al., 2021*) employs a curriculum-meta learning approach to swiftly adapt to new tasks and mitigate interference from previous work.

Relation prototypes CRE (RP-CRE) (*Cui et al., 2021*) employs relation prototypes and multi-head attention mechanisms to address the issue of the hidden representation space being disrupted after learning new tasks.

Consistent representation learning (CRL) (*Zhao et al., 2022*) introduces supervised contrastive learning and knowledge distillation to maintain previously learned knowledge.

Continual relation extraction framework with contrastive learning (CRECL) (*Hu et al., 2022*) consists of a classification network and a contrastive network, decouple the representation and classification process

Classifier decomposition (CDec) (*Xia et al., 2023*) introduces a classifier decomposition framework to address biases in CRE, promoting the learning of robust representations.

## Main results

The overall performance of our proposed model on two datasets is shown in Table 1. All baseline model results mentioned in the table are cited from their respective papers. From the analysis of Table 1, the following conclusions can be drawn:

**Table 1  Accuracy (%) on all observed relations after learning each task.** The best results are marked in bold.

| Dataset | Method | $T_1$ | $T_2$ | $T_3$ | $T_4$ | $T_5$ | $T_6$ | $T_7$ | $T_8$ | $T_9$ | $T_{10}$ |
|---------|--------|------|------|------|------|------|------|------|------|------|------|
| FewRel | EA-EMR | 89.0 | 69.0 | 59.1 | 54.2 | 47.8 | 46.1 | 43.1 | 40.7 | 38.6 | 35.2 |
| | EMAR(BERT) | **98.8** | 89.1 | 89.5 | 85.7 | 83.6 | 84.8 | 79.3 | 80.0 | 77.1 | 73.8 |
| | CML | 91.2 | 74.8 | 68.2 | 58.2 | 53.7 | 50.4 | 47.8 | 44.4 | 43.1 | 39.7 |
| | RP-CRE | 97.9 | 92.7 | 91.6 | 89.2 | 88.4 | 86.8 | 85.1 | 84.1 | 82.2 | 81.5 |
| | CRL | 98.2 | 94.6 | 92.5 | 90.5 | 89.4 | 87.9 | 86.9 | 85.6 | 84.5 | 83.1 |
| | CRECL | 97.8 | 94.9 | 92.7 | 90.9 | 89.4 | 87.5 | 85.7 | 84.6 | 83.6 | 82.7 |
| | CDec | 98.4 | 95.4 | 93.2 | **92.1** | 91.0 | **89.7** | 88.3 | 87.4 | **86.4** | 84.2 |
| | Ours | 98.1 | **95.6** | **94.0** | 92.0 | **91.3** | 89.4 | **88.8** | **87.6** | 86.0 | **84.6** |
| TACRED | EA-EMR | 47.5 | 40.1 | 38.3 | 29.9 | 28.4 | 27.3 | 26.9 | 25.8 | 22.9 | 19.8 |
| | EMAR(BERT) | 96.6 | 85.7 | 81.0 | 78.6 | 73.9 | 72.3 | 71.7 | 72.2 | 72.6 | 71.0 |
| | CML | 57.2 | 51.4 | 41.3 | 39.3 | 35.9 | 28.9 | 27.3 | 26.9 | 24.8 | 23.4 |
| | RP-CRE | 97.6 | 90.6 | 86.1 | 82.4 | 79.8 | 77.2 | 75.1 | 73.7 | 72.4 | 72.4 |
| | CRL | 97.7 | 93.2 | 89.8 | 84.7 | 84.1 | 81.3 | 80.2 | 79.1 | 79.0 | 78.0 |
| | CRECL | 96.6 | 93.1 | 89.7 | 87.8 | 85.6 | **84.3** | **83.6** | 81.4 | 79.3 | 78.5 |
| | CDec | **97.9** | 93.1 | 90.1 | 85.8 | 84.7 | 82.6 | 81.0 | 79.6 | 79.5 | 78.6 |
| | Ours | 97.5 | **94.3** | **91.3** | **88.3** | **85.1** | 83.2 | 82.2 | **81.4** | **80.3** | **79.3** |

(1) Our model achieves competitive results compared to existing baseline models, obtaining the best results in most tasks. This demonstrates the effectiveness of our proposed method in CRE.

(2) Our model did not achieve the best results when evaluating Task 1 on both datasets. This is because, in the initialization phase, there is no need to mitigate catastrophic forgetting through relation prototypes and knowledge distillation. As tasks continue to arrive, maintaining previously learned knowledge becomes more crucial, leading to better performance for our model.

(3) All models perform better on the FewRel dataset compared to the TACRED dataset. This could be attributed to the imbalanced distribution of relation types and the relatively longer sentences in the TACRED dataset, which increases the search space and may lead to performance degradation.

## Ablation study

To validate the impact of the Label Knowledge Infusion (LKI) module and the Multi-Similarity Distillation (MSD) module on the model's performance, we conducted an ablation study, with the results shown in Table 2. Specifically, for w/o LKI (w/o context), We do not randomly insert relation labels into the text context when training on new tasks. For w/o LKI (special token), we replaced relation labels with special tokens. In other words, during the training of a new task, a special token was randomly inserted into each training instance for each relation, and this special token was used to generate relation prototypes through multi-head attention operations. For w/o MSD, we did not employ multi-similarity distillation during the memory replay stage to maintain previously learned knowledge. As for avg. prototype, we obtained relation prototypes by averaging the hidden representations of typical samples. The results from Table 2 indicate that removing specific

**Table 2  Ablation study results.** We remove label knowledge infusion (LKI), Multi-smilarity distillation (MSD) in order and report the accuracy of the last five tasks in FewRel and TACRED. The best results are shown in bold.

|  | Method | $T_6$ | $T_7$ | $T_8$ | $T_9$ | $T_{10}$ |
|---|---|---|---|---|---|---|
|  | Intact Model | **89.4** | **88.8** | **87.6** | **86.0** | **84.6** |
|  | w/o MSD | 87.9 | 86.9 | 85.5 | 84.7 | 84.0 |
| FewRel | w/o LKI (w/o context) | 88.6 | 87.4 | 86.3 | 84.5 | 82.7 |
|  | w/o LKI (special token) | 88.4 | 87.4 | 85.9 | 84.0 | 83.4 |
|  | avg. prototype | 88.0 | 86.9 | 85.4 | 84.1 | 82.4 |
|  | Intact Model | 83.2 | **82.2** | **81.4** | **80.3** | **79.3** |
|  | w/o MSD | 82.4 | 81.1 | 80.2 | 79.8 | 78.6 |
| TACRED | w/o LKI (w/o context) | 83.2 | 81.4 | 79.6 | 79.0 | 78.0 |
|  | w/o LKI (special token) | 82.0 | 81.2 | 80.5 | 79.0 | 78.4 |
|  | Avg. prototype | **83.8** | 81.2 | 80.1 | 79.5 | 77.8 |

modules leads to a decrease in model performance, demonstrating the effectiveness of each module in our model. Not using MSD for knowledge distillation during the memory replay phase disrupts the distribution of relation embeddings in the embedding space, resulting in poorer performance on previously learned relation categories.

To figure out how the label knowledge infusion module work, we conducted the following experiment. Firstly, we employed t-SNE to visualize the distribution of relation labels in the embedding space at different epochs during the learning of new tasks. This aimed to confirm whether relation labels could acquire more semantic knowledge through training *via* textual context. The results are depicted in Fig. 2. During training new task, with an increase in the number of iterations, the embedding of relation labels moves from the boundary of the embedding space toward the center. This indicates that, through self-attention mechanism (*Vaswani et al., 2017*) and gradient propagation, relation labels can capture different patterns in training instances. If this process is not conducted, directly using relation label lacking contextual semantics to generate relation prototypes will result in a significant performance drop. Subsequently, replacing relation labels with special tokens also led to a decrease in model performance. We believe that, compared to using special tokens, relation labels provide a highly generalized semantic understanding of the corresponding relationship. This plays a crucial role both in injecting semantic information into tokens representing relation labels and in generating relation prototypes.

## Analysis of memory size

To mitigate catastrophic forgetting, when new data arrives, we need to select a certain number of instances for memory replay using the k-means algorithm. The selection is limited by the memory size. Usually, memory size is a significant factor influencing model performance. Due to resource constraints and computational limitations, only a small number of representative instances are stored. This requires that the model is minimally affected by changes in memory size. To investigate the performance of our model under different memory sizes, we conducted experiments with memory sizes set to 2, 5, and 15

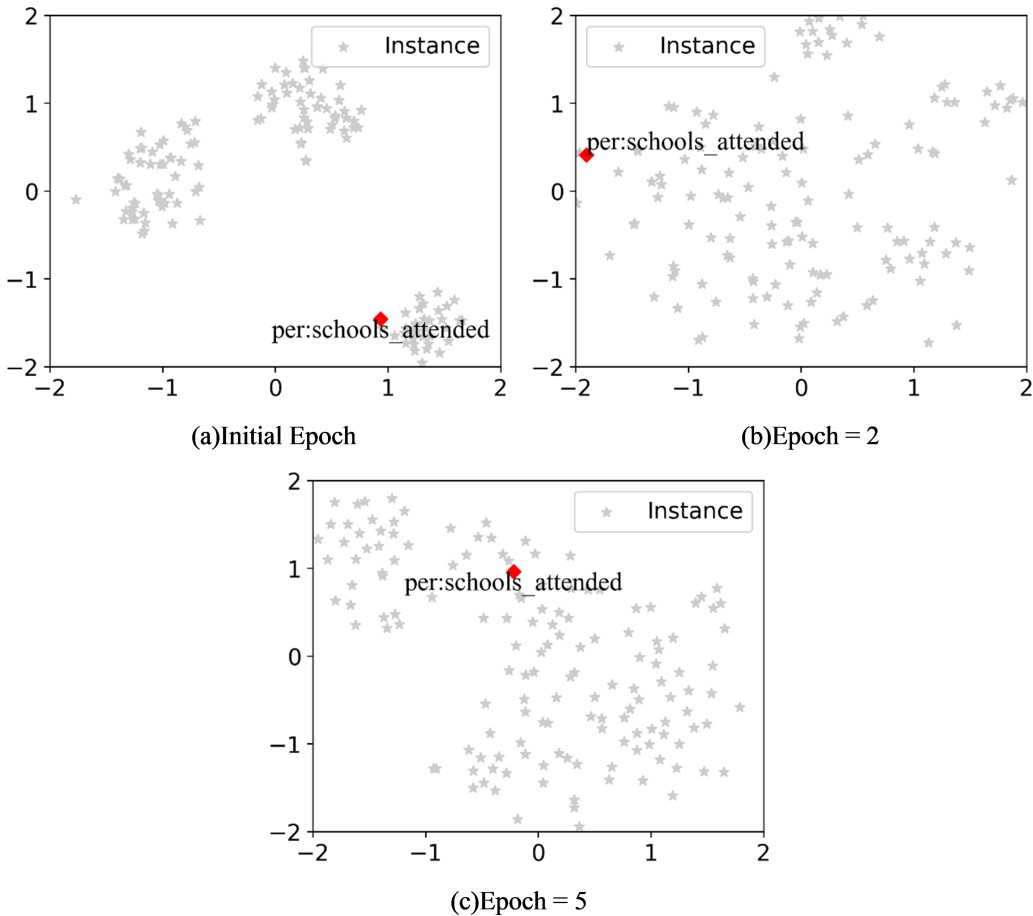

(a)Initial Epoch       (b)Epoch = 2

(c)Epoch = 5

**Figure 2** **(A–C) The distribution of relation labels across different epochs in the embedding space.**

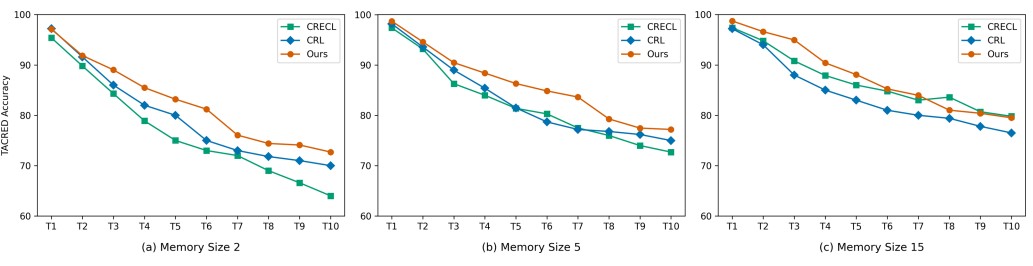

(a) Memory Size 2      (b) Memory Size 5      (c) Memory Size 15

**Figure 3** **(A–C) Accuracy with reference to different memory sizes.**

instances, respectively. We compared the results with baseline models CRL and CRECL. The experimental results are shown in the Fig. 3.

When the memory size is set to 2 and 5, the performance of our proposed model is better than the baseline models. We believe this phenomenon can be attributed to two reasons: (1) During the new task training phase, we inject rich semantic information into the relation labels, acting as a knowledge base. (2) We employ a multi-attention mechanism

**Table 3** **All relations are divided into three groups based on their similarity.** "Accuracy" indicates the average accuracy of the model after learning. "Drop" indicates the accuracy drop after learning.

| Model | Similarity | TACRED | |
| --- | --- | --- | --- |
| | | **Accuracy** | **Drop** |
| | [0.85,1.00) | 64.8 | 11.4 |
| **CRL** | [0.70,0.85) | 76.6 | 5.0 |
| | (0.00,0.70) | 89.6 | 0.6 |
| | [0.85,1.00) | 60.7 | 13.9 |
| **CRECL** | [0.70,0.85) | 70.0 | 8.4 |
| | (0.00,0.70) | 79.9 | 4.3 |
| | [0.85,1.00) | 67.6 | 9.8 |
| **Ours** | [0.70,0.85) | 78.1 | 4.2 |
| | (0.00,0.70) | 88.7 | 1.2 |

to generate relation prototypes, making full use of the limited information contained in the memory.

When the memory size is set to 15, our model did not consistently achieve the best performance in some cases. We believe that with a memory size of 15, the instances stored in memory already provide sufficient information for the model to retain prior knowledge. This results in our proposed method having limited impact on performance improvement.

### Effectiveness of multi-loss distillation

The main reason for catastrophic forgetting in CRE is the presence of similar relations in the learning data stream. Due to their similar semantics, instances of these relations tend to cluster together in the embedding space, leading to classification difficulties. Table 3 shows the performance of CRL and CRECL on different groups of similar relations, where it can be observed that the more similar the learned relations are, the greater the performance decline of the model. Therefore, to improve model performance, it needs to have the ability to distinguish similar relations. Additionally, the proposed model demonstrates a smaller performance decline after learning similar relations compared to the baseline models, proving its stronger capability to distinguish similar relations.

To investigate the impact of multi-loss distillation on the distribution of instances of different relations in the embedding space, we selected five highly similar relations from the TACRED dataset and visualized the distribution of their corresponding samples using t-SNE. From Fig. 4, it can be seen that through multi-loss distillation, instances with the same relation label tend to cluster together in the embedding space, while there is a clear boundary between instances of different relations.

### CONCLUSION

In this paper, we propose a label-guided relation prototype generation method. To capture different emphases of instances stored in memory, we employ a multi-head attention mechanism to generate relation prototypes. We use the embedding of relation label as query vectors to leverage their semantic knowledge. Additionally, we learn patterns in

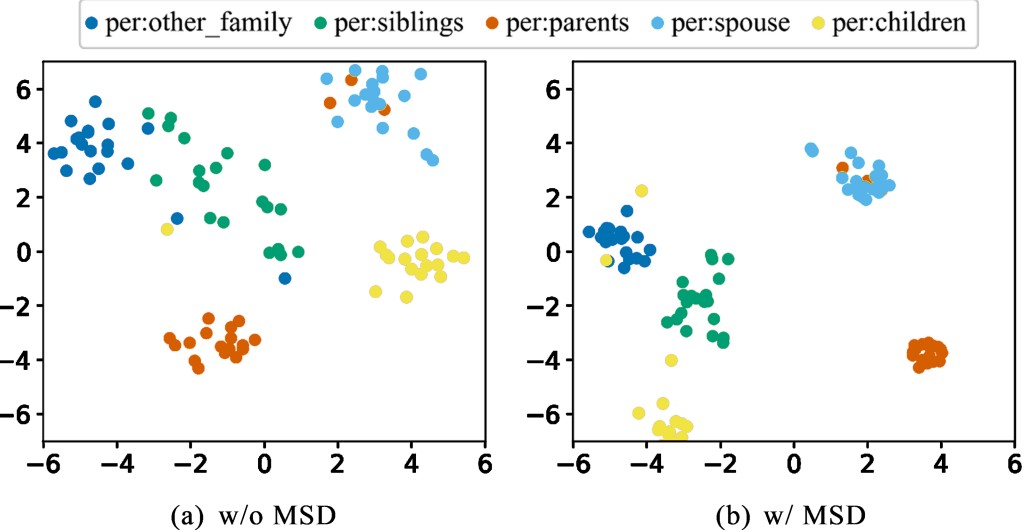

**Figure 4** (A–B) The distribution of five similar relations in the embedding space.

the text by randomly inserting relation labels into the text. Furthermore, we introduce multi-similarity distillation to maintain existing knowledge while making it easier to distinguish between instances with different labels. We validate the effectiveness of the model through a series of experiments and explore how each module work. We believe that the rich information contained in relation labels has not been fully utilized. In future work, we plan to investigate how to full use of these labels, such as expanding the memory by generating pseudo-instances only through entity pairs and relation label embeddings to improve model performance.

## ACKNOWLEDGEMENTS

Thank you to everyone who provided valuable feedback on this article.

### Funding

This article is supported by the 2023 Humanities and Social Sciences Research and Planning Fund of the Ministry of Education with Grant no.23YJA860010. The funders had no role in study design, data collection and analysis, decision to publish, or preparation of the manuscript.

### Grant Disclosures

The following grant information was disclosed by the authors:
The 2023 Humanities and Social Sciences Research and Planning Fund of the Ministry of Education: 23YJA860010.

## Competing Interests

The authors declare there are no competing interests.

## Author Contributions

- Shuang Liu performed the experiments, authored or reviewed drafts of the article, and approved the final draft.
- XunQin Chen conceived and designed the experiments, prepared figures and/or tables, and approved the final draft.
- Peng Chen analyzed the data, prepared figures and/or tables, and approved the final draft.
- Simon Kolmanič performed the computation work, authored or reviewed drafts of the article, and approved the final draft.

## Data Availability

The source code are available in the Supplementary File.

The fewrel dataset is available at GitHub: https://github.com/thunlp/FewRel.

The TACRED dataset is available at https://catalog.ldc.upenn.edu/LDC2018T24.

## Supplemental Information

Supplemental information for this article can be found online at http://dx.doi.org/10.7717/peerj-cs.2327#supplemental-information.

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
