# Peer review of "Label-Guided relation prototype generation for Continual Relation Extraction"

_PeerJ Computer Science, doi:10.7717/peerj-cs.2327_

## Round 0.1 · original submission · Major Revisions

Please find appended the detailed comments from the reviewers and revise the work accordingly. Then it will be evaluated again.

·

Basic reporting

The paper “Lable-Guided Relation Prototype Generation for Continual Relation Extraction” by Shuang Liu et al. is devoted to a new method for Continual Relation Extraction.
Proposed method extracts semantic information from the label namespace, resulting in adequate descriptions of relation semantics by relation prototypes. As follows from the abstract, the method allows one to relatively well overcome the problem of catastrophic forgetting. According to the authors, the advantage of the method is also the fact that extracting additional information from data is not limited by memory capacity. The stated general description of the algorithm in the Abstract sounds very interesting.

The logic of presentation suggests the approximately following structure for such an article: first (1) a description of existing methods, then (2) the theory used, (3) the developed algorithm, (4) experiment, (5) analysis of the results of applying the method, (6) comparison of the results of the method with the results of the methods discussed at the beginning, (7) conclusion. This plan makes it possible to explain the material step by step. Formally, the authors adhered to almost this scheme. Let's look at all the parts.

The first step ((1) a description of existing methods) in the paper actually presents in parts Introduction and Related Work (“Work” is written in the singular for some reason).

Then the second and third logical steps ((2) the theory and (3) the developed algorithm) are mixed in one part Methodology. As a result, the description of the developed algorithm is simply lost in the description of the theory. It is proposed to follow the plan and first describe the theoretical part, and then the algorithm of the method (in particular describe all the symbols in Figure 1, which, as I understand it, depicts the algorithm).

The next three logical steps ((4) experiment, (5) analysis of the results of applying the method and (6) comparison of the results of the method with the results of the methods discussed at the beginning) are presented in one part Experiment with references to Table I, Table II, Figure 2, Figure 3 and Figure 4. The result of mixing several topics in this part is compounded by the level of evidence. The authors make a strong statement that “method has achieved competitive performance compared to the state-of-the-are baseline models in CRE”, but in Table I does not even average the data across all Tasks, which will make it possible to evaluate of the effectiveness of the developed method. The Ablation Study section describes the results of the experiments without explaining which part of the algorithm the abbreviations used apply to (LKI on lines 232, 233 and MSD on lines 236, 239). The Memory Size Analysis section describes the memory size in unnamed units (line 260). In the section Effectiveness of Multi-Loss Distillation the results are explained based on two scatterplots (on Figure 4) without a clear description. And, unfortunately, there is no analysis of how the method allows to overcome the problem of catastrophic forgetting, as indicated in the Abstract. It is proposed to rewrite the part according to the plan, increase the level of evidence taking into account comments and make corrections to the Supplement.

Minor comments:
1) In formulas:
- in the formula (1) it is unclear what W_1 [h^11;h^21 ] is, it does not explain what d is in R^d (line 125).
- in the formula (2) there is W_2, but after the formula W_1 is described (line 128) and h is not described.
- after the formula (3) the description of P(r|x_i;θ_k ) is unclear (line 130).
- in the formula (5) the undescribed operator Att. and d_1 are used, linear transformation of H_(x_r ) isn’t described.
- after the formula (7) z_(x_i ),z_(r_i ),z_r,τ_1 are not described.
- in the formula (9) the variable S_im should probably have other indices - S_ij.
- in the formulas (10) and (11) it is also necessary to check the indices.
- after the formulas (12) and (13) the variables α,β,λ are not described.
- in the formula (15) L_rd is used, but in the formula (8) L_nd is used. Are these different variables?
2) Why make Continual Learning section in the Related Work part if there is only one?
3) Unclear sense of sentences on lines 130, 131, 151, 169.
4) The text contains more than 40 obvious spelling errors, even the title of the article contains an error.
5) Captions on Figures and Tables are unclear. The scales in the scatterplots in Figure 2 are different, as are those in Figure 4.

Overall, the article presents a potentially interesting method. However, the structure of the paper and some presented results are somewhat confusing and require either more explanation or corrections.

Experimental design

No comment

Validity of the findings

No comment

Reviewer 2 ·

Basic reporting

The paper is written in a professional tone and the structure of the paper is clear, with some typos throughout.
Key references are provided for baselines and datasets.
Figures and tables are produced in a professional way, the source code of the model is provided for reproducibility, but the raw data is not.
Results and ablations studies can support the claims of the paper.
The problem formulation and the notation are clear.

Experimental design

The experiment design is standard for the continual relation extraction problem.
However, there is inherent randomness in the results due to the random split of the relations for the ten tasks. The baseline results are directly taken from the respective paper, instead of from running the algorithms on the same split as the proposed method. This would make the comparison less convincing. A more rigorous setting would be to reproduce the baseline results and compare on the same split. Additionally, it would be beneficial to report the variance of the accuracy numbers and performance significance tests.

Model hyperparameters (such as the selection criteria for the positive and negative examples in the multi-similarity loss) are not reported in the paper but can be found the supplementary code.

Validity of the findings

For model design, since the task is to classify instances into relation types, including the relation label name in the input would be directly leaking the label. A more reasonable way of utilizing the label information could be using label names to initialize the classification layer (W_2 parameters).

Unlike CRL which utilizes the supervised contrastive learning objective for both the new task and the experience replay, this paper proposes to use cross-entropy loss for the new task and supervised contrastive learning with the relation prototype for replay. This causes some discrepancies in learning a relation. Although the authors have shown experiment results, I would like to see some justification for this design choice.

Finally, the validity of the experiment results could be undermined by the randomness of the relation split.

Additional comments

I would suggest the authors do a grammar check on their manuscript, here's a non-comprehensive list of writing errors to fix:
- Line 14: "traing" should be "training"
- Line 21: "enabling the leverage of .." -> "leveraging"
- Line 25: "state-of-the-art"

- Line 36: "constraintes" -> "constraints"
- Line 40: "relation prototypes calculation is relatively coarse" -> "relatively simplistic"
- Line 47: "existing memory-babsed methods may encounter follow limitations" -> "memory-based methods encounter the following limitations"
- Line 53: "emloy" -> "employ"
- Line 54: "approch" -> "approach"
- Throughout the paper there are many mentions of "representative examples" for old relations. In the introduction they are referred to as 'typical examples' and in the problem formulation they are referred to as "classical examples". Please use a unified name, either "representation examples", "typical examples" or "examplars" would be good.

- Line 128: W_1 should be W_2 instead.
- Line 139: "utilize the hidden representation of the relation label as a probe" -> "relation label as the query vector"?

Cite this review as

---

## Round 0.2 · accepted · Accept

The two prior reviewers were not available to re-review, however, I have checked the manuscript and to the best of my knowledge, I believe the authors addressed the issues proposed by the reviewers.